# Application and Effect of *Pediococcus pentosaceus* and *Lactiplantibacillus plantarum* as Starter Cultures on Bacterial Communities and Volatile Flavor Compounds of *Zhayu*, a Chinese Traditional Fermented Fish Product

**DOI:** 10.3390/foods12091768

**Published:** 2023-04-24

**Authors:** Dongmei Xu, Yongle Liu, Xianghong Li, Faxiang Wang, Yiqun Huang, Xiayin Ma

**Affiliations:** 1School of Food and Biological Engineering, Changsha University of Science & Technology, Changsha 410114, China; 2Hunan Provincial Engineering Technology Research Center of Aquatic Food Resources Processing, School of Food Science and Bioengineering, Changsha University of Science & Technology, Changsha 410114, China

**Keywords:** *Zhayu*, LAB fermentation, microbial communities, texture, volatile flavor compounds

## Abstract

*Zhayu* is a type of traditional fermented fish product in China that is made through the fermentation of salted fish with a mixture of cereals and spices. Inoculation fermentation was performed using *Pediococcus pentosaceus* P1, *Lactiplantibacillus plantarum* L1, and a mixture of two strains, which were isolated from cured fish in Hunan Province. Compared with the natural fermentation, inoculation with lactic acid bacteria (LAB) accelerated the degradation of myosin and actin in *Zhayu*, increased the trichloroacetic acid (TCA)-soluble peptide content by about 1.3-fold, reduced the colony counts of *Enterobacteriaceae* and *Staphylococcus aureus* by about 40%, and inhibited their lipid oxidation. In the texture profile analysis performed, higher levels of hardness and chewiness were observed in the inoculation groups. In this study, the bacterial community and volatile flavor compounds were detected through 16S high-throughput sequencing and headspace solid-phase microextraction–gas chromatography–mass spectrometry (HS-SPME-GC-MS). Inoculation with *L. plantarum* L1 reduced around 75% abundance of *Klebsiella* compared with the natural fermentation group, which was positively correlated with 2,3-Butanediol, resulting in a less pungent alcohol odor in *Zhayu* products. The abundances of 2-pentylfuran and 2-butyl-3-methylpyrazine were increased over threefold in the L1 group, which may give *Zhayu* its unique flavor and aroma.

## 1. Introduction

Fermentation has been widely used in fish processing in order to prolong the fish products’ shelf lives and develop unique flavors [1]. In China, there are a variety of traditional fermented fish products produced in ethnic minority areas, such as *Chouguiyu* [1,2], *Suanyu* [3], and *Yucha* [4]. *Zhayu* is a type of solid fermented fish product from Hunan Province, China. In the traditional natural fermentation of *Zhayu*, washed fish fillets were cubed and combined with salt, rice flour, cayenne, and other seasonings, and after that, the mixture was pressured and cultured in a solid form [5]. However, along with globalization and market opening, natural fermented *Zhayu* products face many difficulties in achieving large-scale industrial production. The quality of *Zhayu* products is influenced by a variety of factors, such as environmental microorganisms, seasonal variation, and manual operation experience [6,7], leading to long fermentation cycles and the high variability of product quality.

According to relevant reports, inoculation fermentation is a frequently employed technique for raising the quality of fermented fish products, and starter cultures used for fermentation are commonly selected from *Lactobacillus*, *Pediococcus*, *Staphylococcus*, *Saccharomyces,* and related genera [8,9,10]. The physiological functions and characteristics of LAB were crucial to the fermentation process. Spoilage microorganisms could be inhibited by antibacterial metabolites produced by LAB, including organic acids and bacteriocins [11,12]. Studies have proved that LAB could encourage the development of fish products with improved flavors [13].

The flavor of fermented foods was correlated with the microbial composition during fermentation [14]. High-throughput sequencing based on the 16S rRNA gene amplicon has been utilized to research microbial communities in fermented fish products [15,16,17]. The relationship between microbial succession and flavor development has been investigated in *Suanyu* inoculated with diverse starter cultures [18]. A further study investigated how diverse cultures affected the characteristic flavor of *Suanyu* [19]. In Chinese fish sauce, Wang et al. [20] investigated the dynamics of volatile flavor chemicals and their association with microbial populations. As for research on *Zhayu*, Yang et al. [8] found that the addition of flavorzyme could enhance the synthesis of alcohols, aldehydes, and esters in *Suanzhayu*. An et al. [5] found that *Zhayu* inoculated with *L. plantarum* and *P. acidilactici* showed rising acidity and a pleasant odor, and the amount of terpenoids, esters, acids, and S-containing compounds were increased in *Zhayu*. However, the relationship between flavor quality and microbial composition remains to be exploited, and strains with good fermenting properties need to be identified.

Previously, two strains of LAB (*P. pentosaceus* P1 and *L. plantarum* L1) were chosen from the traditional natural fermentation of cured fish and identified as having good fermentation properties, such as salt tolerance and antibacterial ability. In the current investigation, fermented *Zhayu* samples were generated by inoculation with *L. plantarum*, *P. pentosaceus*, or a mixture of the two strains. The safety and textural properties of *Zhayu* were evaluated, and the bacterial communities and volatile flavor compounds of *Zhayu* were investigated. This study aimed to ascertain the impact of inoculation with LAB species on the quality of *Zhayu* and provide information on the relevance between microbial variation and flavor enhancement.

## 2. Materials and Methods

### 2.1. Materials and Chemicals

The grass carp (*Ctenopharyngodon idellus*) and rice flour were bought from a partial market in Changsha City, Hunan Province. Red yeast rice flour (Jiangsu Weipinhui Food Co., Ltd. Taizhou, China) was purchased from the supermarket of Changsha University of Technology. The molecular weight marker used was the protein ladder (range 10–200 kDa, Thermo Fisher Scientific Inc. MA, USA). The de Man, Rogosa, and Sharpe (MRS) medium, Violet Red Bile Glucose Agar (VRBA), and Mannitol Salt Agar (MSA) were bought from Huankai Microbial Co., Ltd. (Guangzhou, China). All additional chemicals used were of analytical quality.

### 2.2. Pre-Culture of LAB

*P. pentosaceus* P1 and *L. plantarum* L1 were isolated from cured fish and preserved in the laboratory of the School of Food and Biological Engineering, Changsha University of Science and Technology. The LAB strains were revitalized twice in MRS broth at 37 °C for 12 h. Then, culture solutions were centrifuged at 4 °C for 10 min with a centrifugal force of 11,100× *g*. Then, the bacterial cell pellets were re-suspended in sterile saline water and adjusted to a cell density of 10^6^ CFU/mL.

### 2.3. Preparation of Zhayu Samples

Fresh grass carps (weight: 2.5 ± 0.5 kg) were slaughtered and stored in crushed ice immediately. White dorsal muscle was removed and processed into blocks of 3 × 3 × 2 cm^3^. The grass carp pieces were immersed in brine solution (1:25 g/g, salt to fish; 1:10 *w/v*, salt to water) at 4 °C for 24 h and then patted dry with paper towels. Following this, the grass carp pieces were patted with paper and dried for 2 h at 45 °C in an electric blast drying oven (DHG-9140A, Shanghai Jinghong Instruments Co., Ltd., Shanghai, China). Based on our previous study, the carp pieces were mixed with 20% (g/g) rice flour, 4% (g/g) red yeast rice flour, and 3% (*w/v*) LAB starter cultures (Figure 1). Samples were fermented in 50 mL jars with lids sealed tightly at 32 °C in a constant temperature incubator (LHS-250SC, Shanghai Yiheng Technology Co., Ltd., Shanghai, China) and named CK (without starter cultures), P1 (inoculated with *P. pentosaceus* P1), L1 (inoculated with *L. plantarum* L1), and PL (inoculated with the 1:1 mixture of *P. pentosaceus* P1 and *L. plantarum* L1). As shown in Figure 1. The samples were taken from separate containers at 0, 2, 4, 6, 8, and 10 d during the fermentation process for inspection. At each sampling location, three parallel samples were collected for analysis.

### 2.4. Determination of pH and Trichloroacetic Acid (TCA)-Soluble Peptides

The pH values of *Zhayu* were measured according to the national standard GB 5009.237-2016 [21]. Samples (1 g) were homogenized with 10 mL of 0.75% (*w/v*) KCl solution and then measured with an electronic pH meter (Ohaus International Trading Co., Ltd., Shanghai, China).

According to the method of Hatairad et al. [22], the TCA-soluble peptide contents of *Zhayu* products during fermentation were examined. Samples (2.0 g) were homogenized with 18 mL of 5% (*w/v*) TCA, and the mixture was extracted at 4 °C for 1 h while being vibrated every 15 min. The solution was centrifuged at 4000× *g* and 4 °C for 15 min, and then the supernatant was collected and mixed with Folin reagent to produce a dark blue complex. The TCA-soluble peptides were observed at 500 nm using the UV spectrophotometer (TU-1901, Beijing Pu-Analysis General Instrument Co., Ltd., Beijing, China). The results were reported as µmol tyrosine/g sample. Each determination was made three times.

### 2.5. Determination of Microbial Counts

Five grams of each sample was aseptically removed from the jars and homogenized for 1 min in 45 mL of 0.9% (*w/v*) saline solution. A series of gradient dilutions were prepared, and 100 μL of bacterial suspensions was dispersed on plates for microbial counting. LAB was cultured on MRS medium at 37 °C for 24 h, *Enterobacteriaceae* were cultured on VRBA at 37 °C for 24 h, and *Staphylococcus aureus* strains were cultured on MSA at 37 °C for 24 h. The number of colonies was expressed as log colony forming units (CFU) per gram of *Zhayu* sample.

### 2.6. SDS-Polyacrylamide Gel Electrophoresis (SDS-PAGE)

The SDS-PAGE procedure is modified from Li et al. [23] in certain ways. Briefly, the extracted protein solutions were adjusted to the same concentration with 1× loading buffer solution and then heated at 100 °C for 5 min prior to electrophoresis. The molecular weight marker used was protein ladder (range 10–200 kDa). The concentrated gel and separated gel were 5 g/mL and 120 g/mL. The electrophoresis equipment (DYCZ-25D, Beijing Liuyi Biotechnology Co., Ltd., Beijing, China) was run at a continuous current of 20 mA for the concentrated gel and 40 mA for the separated gel, which had concentrations of 5 g/mL and 120 g/mL. They were stained with Coomassie brilliant blue R250, followed by destaining to visualize the gels. Results from scanning were obtained utilizing a Bio-5000 Plus GEL imaging scanner (Shanghai Microtek Technology Co., Ltd. Shanghai, China).

### 2.7. Determination of Thiobarbituric Acid-Reactive Substances (TBARS)

Two-gram portions of *Zhayu* samples were combined with 10 mL of trichloroacetic acid solution (75 g/L trichloroacetic acid, 1 g/L propylgallate, and 1 g/L EDTA). The homogenate was filtered through double-layer filter paper, and 5 mL of the filtrate was combined with 5 mL of the 0.02 mol/L 2-thiobarbituric acid solution and heated in boiling water (90 °C) for 30 min. After cooling with ice water, the mixture was analyzed using an ultraviolet spectrophotometer (Beijing General Instrument Co., Ltd. Beijing, China) at 532 nm [24]. The TBARS values were expressed as mg of malondialdehyde (MDA) per kg of sample.

### 2.8. Texture Profile Analysis (TPA)

Texture profile analysis (TPA) of *Zhayu* samples (1.5 cm radius and 2.0 cm in thickness) was performed using the Texture Analyzer TA.XT.plus (Stable Micro Systems Ltd., Godalming, UK). According to Wang et al. [13], the following experimental parameters were chosen: a trigger force of 0.049 N, a compression ratio of 30%, a pre-test speed of 1.0 mm/s, a test speed of 1.0 mm/s, and a post-test speed of 2.0 mm/s. Using a cylinder probe P/50, six parallel experiments were conducted before each test.

### 2.9. Low-Field Nuclear Magnetic Resonance (LF-NMR)

Samples were cut into 10 mm × 10 mm × 10 mm (1 g) cubes along the fiber direction and placed in nuclear magnetic resonance tubes (25 mm in diameter). The transverse relaxation data were measured using a MesoMR23-060V-I LFNMR analyzer (Niumag, Ltd., Shanghai, China) according to the method of Qin et al. [25]. The analyzer was operated at a resonance frequency of 18 MHz at 32 °C. The Carr–Purcell–Meiboom–Gill (CPMG) pulse sequence was used to measure the T2. The time delay used for the T2 measurement was between 90 °C and 180 °C, with pulses of 14 ms and 24 ms. There were 200 measurement points for the CPMG measurements. Data from 4000 echoes were collected over the course of 8 scan repeats, with a 9000 ms repeat interval between each scan. The multi-exponential decay curve was obtained from NMR relaxation processing by conducting multi-exponential fitting analysis with Nuimag’s Multi Exp Inv Analysis software.

### 2.10. Analysis of Bacterial Diversity

Samples were obtained in a sterile environment and kept at −80 °C. The entire genome of DNA in the samples was extracted using the CTAB/SDS method. In 1% (*w/v*) agarose gels, DNA concentration and purity were examined. DNA was diluted to a concentration of 1 ng/L using sterile water.

The 16S rRNA genes were amplified in different locations (16S V3-V4) using a particular primer (16S V4: 515F-806R) and barcodes. Amplifications were performed employing an initial denaturation cycle at 98 °C for 1 min, followed by 30 cycles of denaturation (98 °C for 10 s), annealing (50 °C for 30 s), elongation (72 °C for 30 s), and a final 5 min extension at 72 °C. All PCR mixtures contained 15 L of Phusion^®^ High-Fidelity PCR Master Mix (New England Biolabs, London, UK), 0.2 µM of each primer, and 10 ng of target DNA. PCR products were resolved by a Qiagen Gel Extraction Kit (Dusseldorf, Germany). Using the NEBNext^®^ UltraTM IIDNA Library Prep Kit, sequencing libraries were created (Cat No. E7645) and assessed using the Agilent Bioanalyzer 2100 system and the Qubit@ 2.0 Fluorometer from Thermo Scientific (Waltham, MA, USA). Ultimately, 250 bp paired-end reads were produced after the library was sequenced on an Illumina NovaSeq device. Sequences were combined using FLASH and quality filtered before being grouped into operational taxonomic units (OTUs) by UPARSE at a 97% similarity threshold, and chimaeras were detected using UCHIME.

### 2.11. Determination of Volatile Compounds by HS-SPME-GC-MS Analysis

Each sample (5 g) was chopped and put into a headspace vial with a 20 mL capacity and then placed in a robotic arm TriPlus RSH (CTC Analytics, Basel, Switzerland) autosampler incubation chamber and heated at 40 °C for 10 min for volatile compound enrichment. The extraction was performed using a headspace solid-phase microextraction (HS-SPME) extraction needle (50/30 μm DVB/CAR/PDMS Supelco, MA, USA) for 20 min. Using the GC-Orbitrap-MS system (Q Exactive GC, Thermo Fisher, MA, USA), the volatile chemicals were examined on a TG-5 column (0.25 × 0.25 × 0.32, 30 m, Fisher Scientific, MA, USA).

The GC-MS started out at 40 °C for 2 min and was then raised to 250 °C at a rate of 8 °C/min and held for 2 min, for a total running cycle of 32 min. The MS was run in positive electron impact ionization EI+ mode with a solvent delay for the first 0.5 min, scanning from ion mass fragments 35–475 *m/z*, an interscan delay of 0.1 s, and a resolution of 60,000 at FWHM (Full Width at Half Maximum). The carrier gas was helium of 99.996% purity (Shandong, China). The velocity of the helium gas was set at 1.2 mL/min.

For the retention times and retention index of volatile flavor compounds, we relied on computer matching with the reference of the MS library of NIST 14.0 (mass-spectral similarity match ≥ 80). By computing the percentages of GC peak regions, the compounds were given a numerical value.

### 2.12. Data Analysis

For each experiment, at least three technical replicates were examined using the statistical program SPSS 26.0. The results were presented as mean standard deviation (SD). Using Welch’s ANOVA (α = 0.05), the Dunnet-T3 post-hoc test was used to examine the significance of the data; a probability value of *p* < 0.05 was deemed significant. SIMCA-P 14.0 software (Umetrics, Umeå, Sweden) was used to perform orthonormal partial least squares-discriminant analysis (OPLS-DA) on GC-MS metabolomic data. Characteristic metabolites were defined as variables in the projection, with variable importance (VIP) > 1. GraphPad Prism 8.0.2 was used to examine the significant differences and Pearson correlation coefficient and plot all heatmap construction.

## 3. Results and Discussion

### 3.1. Growth of LAB Counts and Acid Production

As shown in Figure 2a, the pH of naturally fermented *Zhayu* (CK group) dropped rapidly from 6.49 to 5.22 in the first two days and then decreased to 4.25 gradually in the next 8 days. Compared with the CK group, the pH of *Zhayu* inoculated with LAB decreased faster, especially in the L1 group, which reached 4.28 after two days of fermentation. However, there was no significant difference between the P1, L1, and PL groups and only a minor difference in the CK group after 10 days of fermentation. Correspondingly, the growth of LAB in samples is shown in Figure 2b. The LAB counts of the P1, L1, and PL groups were initially higher than those of the CK group and reached above 10^7^ cfu/g at 4 days of fermentation, whereas the LAB counts of the natural fermentation group only reached about 10^6^ cfu/g at 4 days of fermentation.

Although the pH dropped to around 4.0 after 10 days of fermentation in all groups, the inoculation with LAB greatly accelerated the production of lactic acid and significantly reduced the time needed for the fermentation. As indicated in the results, the pH variations of all samples showed the same trend throughout fermentation, which declined first and then tended to stay stable as fermentation time rose. The grass carp sample’s initial pH level was 6.5 ± 0.05. Fermentation took place for 4 days, during which time all samples’ pH changed considerably decreased. It was interesting to see that the inoculation group’s pH was much lower than that of the natural fermentation group, indicating that the early stages of fermentation with *P. pentosaceus* and *L. plantarum* would result in a faster rate of pH lowering for the grass carp samples. In particular, group L1 has the lowest pH. As has been described in another study [19], *L. plantarum* exhibits a better capacity for acid production.

### 3.2. Indicators Related to Protein Degradation

The SDS-PAGE experiment was performed to investigate the protein hydrolysis of *Zhayu*. As shown in Figure 3a, the myosin heavy chain protein (about 200 kDa), α-actinin (about 100 kDa), and actin (about 43 kDa) were significantly hydrolyzed in four groups after two days of fermentation. New protein bands with molecular weights of 120 kDa, 60–85 kDa, 30 kDa, and 15–20 kDa were observed and, in the 4 days of fermentation, all protein bands became noticeably lighter, especially in groups L1 and PL. These results indicated that large proteins, such as sarcoplasmic and myofibrillar proteins, were gradually degraded into small proteins and peptides during the fermentation process, and the content of *L. plantarum* L1 (L1 and PL) could promote protein degradation. The degradation of proteins can be due to the activation of endogenous enzymes and microbial proteases [11,18]. LAB intensified the hydrolysis of muscle proteins and had a strain-dependent effect on the breakdown of myofibrillar and sarcoplasmic proteins [26,27]. Denaturation due to the reduced pH resulted in the three-dimensional structural modification of proteins. Proteins are more easily degraded after denaturation and are more easily attached by proteolytic enzymes than they are in their original condition [28].

A change in TCA-soluble peptides during fermentation was observed. As shown in Figure 3b, the level of TCA-soluble peptides increased from 74.6 μmol/g to 861.5 μmol/g at 10 days of fermentation in the CK group, from 79.7 μmol/g to 995.9 μmol/g in the P1 group of inoculation fermentation, from 83.4 μmol/g to 1037.5 μmol/g in the L1 group of inoculation fermentation, and from 74.6 μmol/g to 1051.6 μmol/g in the PL group of inoculation fermentation. Some authors have reported similar outcomes [26,29]. During fermentation, there was significant hydrolysis of muscle proteins, as evidenced by the rising TCA-soluble peptide level [29,30]. The myofibrillar profiles of all inoculations of fermentation groups were indistinguishable, but the proteolysis was more pronounced in the *L. plantarum*-inoculated *Zhayu* samples. These outcomes matched the ongoing breakdown of proteins, suggesting that inoculation with *L. plantarum* L1 could promote the hydrolysis of proteins.

### 3.3. Improvement of Safety Qualities by LAB Strains

The quantities of *Enterobacteriaceae* and *Staphylococcus aureus* were measured during the fermentation of *Zhayu*. As shown in Figure 4a, although the number of *Enterobacteriaceae* in the CK group decreased gradually during fermentation, there were still 2.06 log CFU/g of *Enterobacteriaceae* in the naturally fermented *Zhayu* after 10 days of fermentation. However, in groups inoculated with LAB, the quantities of *Enterobacteriaceae* were significantly reduced during the fermentation, and we could not detect colonies in the undiluted stock solution of the samples after ten days of fermentation (<1 log CFU/g). Similar results have been observed in the growth of *Staphylococcus aureus* (Figure 4b). The number of *Staphylococcus aureus* in the CK group had a rising inflection point at 2 days of fermentation, which was probably caused by the rich nutrition and oxygen in the initial stage. However, the growth of *Staphylococcus aureus* was inhibited by adding LAB, and *L. plantarum* L1 exhibited the best performance in antibacterial capability, as the quantity of *Staphylococcus aureus* decreased to 1.84 log CFU/g at 8 days of fermentation and could not be detected at 10 days of fermentation.

*Enterobacteriaceae* and *Staphylococcus aureus* are two microorganisms that are frequently linked to food deterioration. Food poisoning can result from *Enterococcus* spp., one of the major pathogens in meat products [31], and it can affect the flavor and safety of food and cause food poisoning. It had been established that LAB, as the dominant bacteria, played an essential function in fermented foods by consuming carbohydrates and proteins to make lactic acid and bacteriocins and limiting the growth of spoilage bacteria, which had a favorable impact on the quality of items [1].

TBARS are frequently used to determine the level of lipid oxidation. As shown in Figure 4c, the level of TBARS in the CK group increased rapidly from 0.5 mg/kg to 2.03 mg/kg in 10 days of fermentation, but only reached about 1.26 mg/kg in the inoculated groups. Among them, the L1 group is significantly better. The TBARS contents of all the *Zhayu* products were below 5 mg/kg, which is the standard discussed in [27]. This suggests that lipid oxidation was within acceptable ranges for the majority of samples, in line with a prior finding [32]. It was claimed that some bacteria, particularly LAB, which had significant proteolytic activities and generated specific dietary protein-derived active peptides with protective potential against lipid oxidation, would limit lipid oxidation during fermentation [12,33,34].

### 3.4. LAB Strain Influence on the Texture Properties

The texture properties of fermented *Zhayu* products are shown in Figure 5. The hardness, adhesiveness, and chewiness of fish products increased after 4 days of fermentation, and these values were higher in the inoculated groups. There was no obvious difference in the elasticity, cohesiveness, and recovery of *Zhayu* at 0 and 4 days of fermentation. There was no significant change in the natural fermentation group at 0 and 4 days of fermentation, but at 4 days of fermentation, these values in the inoculated group were slightly lower than those in the natural fermentation group. The results of the TPA test show that inoculation fermentation improves the quality of the product and makes the fermented fish preserved products more appetizing, which is in agreement with Wang et al.’s [13] observations that, due to the salt infiltration and moisture loss, the muscle became strengthened, resulting in a reduced water-holding capacity and improved chewability. In addition, muscle proteins were acid-denaturated and started to coagulate as a result of the pH drop. During fermentation, gelatinization rises, most likely as a result of protein hydrolysis, an increase in hydrophilic groups, and an increase in the water-holding capacity of fish muscle proteins [28].

T2b (0.1~1 ms) and T21 (1~10 ms) are two of the times when water is strongly attached to the macromolecule and unaffected by atmospheric pressure. Between the membrane and myogenic and reflexogenic fibers, there is fixed water at T22 (10~100 ms); T23 (100~1000 ms) free water does not exist. The various component relaxation times may indicate the composite states of different waters in the fish [3]. The L1 group’s T21 and T22 values decreased in Figure 6, which suggests that more water was bound during the fermentation process, tightening the muscles’ bonds to one another. The absence of a distinctive T2 peak between 100 and 1000 ms before and after fermentation suggests that free water is removed during curing and pre-drying. The three groups that were inoculated with various strains did not significantly differ from one another, whereas the uninoculated and inoculated groups did significantly differ from one another.

### 3.5. Abundance and Diversity of Members of the Bacterial Communities

From the fourth day, 16S rRNA gene sequencing was performed with *Zhayu* samples in order to explore the influence of fermentation with LAB on the bacterial composition. As shown in Table 1, the naturally fermented *Zhayu* had the highest species abundance, whereas the products fermented with mixed starters (PL) had the lowest species abundance. Sequence coverage in all groups of samples was more than 99.0%, indicating that the microbial communities in all samples could be accurately described by the sequencing depth data.

At the genus level (Figure 7a and Figure 8), there were 11 microbial genera with a relative abundance over 1%. *Pantoea* (22.59%), *Klebsiella* (18.77%), *Pediococcus* (17.03%), *Cronobacter* (6.08%), *Kosakonia* (3.54%), *Bacteroides* (3.37%), and *Lactiplantibacillus* (3.11%) were the main bacterial communities in the naturally fermented *Zhayu*. The prevailing genus of bacteria significantly changed after inoculation with LAB strains. In the P1 group, the abundance of *Pediococcus* reached 80.30% after 4 days of fermentation, and *Pantoea* (4.87%), Streptococcus (2.69%), and *Klebsiella* (1.06%) were significantly inhibited. In the L1 group, *Lactiplantibacillus* accounted for 72.30% of the total. The abundances of *Pantoea* (8.91%) and *Klebsiella* were inhibited at 8.91% and 5.08%. In the PL group, the first dominant genus was *Pediococcus* (47.76%), followed by *Lactiplantibacillus* (19.80%), and the other genera with more than 1% were *Pantoea* (14.90%), *Klebsiella* (4.01%), and *Kosakonia* (1.54%). It was found that both the inoculated *P. pentosaceus* and *L. plantarum* strains can thrive in the fermented fish environment and establish themselves as dominant. However, inoculated *P. pentosaceus* strains have a greater competitive advantage over spoilage microorganisms, which may be because they are better suited to grow in fermented fish [3], as evidenced by the fact that *P. pentosaceus* are more plentiful than *L. plantarum* in the natural fermentation group.

According to the findings of the microbiological investigation, the inoculation treatment improved the bacterial quality in the *Zhayu* by boosting the competitiveness of the dominant bacteria (*P. pentosaceus* and *L. plantarum*) to compete and limiting the growth of undesirable microorganisms (spoilage bacteria and pathogens).

We can find high species richness in the natural fermentation group. This is likely because there were no dominant strains present during the early stages of the natural fermentation process, which allowed for the growth of a large number of miscellaneous bacteria. Lactic acid steadily built as fermentation progressed, and the relative abundance of acid-tolerant LAB eventually took control [35]. LAB are helpful microbes for fermentation because they may produce vitamins, minerals, and physiologically active peptides such as proteinase [13,33]. In the fermentation of fish products, *Lactiplantibacillus* is thought to be a key player in the lipolysis and proteolysis of proteins [36]. Several spoilage-related microbes, including *Pantoea*, *Klebsiella*, and *Cronobacter,* were also suppressed in the inoculated fermentation groups. It is possible for *Lactobacillus* to create lactic acid, which lowers pH and prevents the growth of spoilage bacteria [37].

Beta diversity represents a comparison of microbial community composition, assessing differences among microbial communities. Figure 7b depicts the PCoA analysis between the samples, with PC1 accounting for 52.49% of the first principal component and PC2 accounting for 31.47% of the second principal component; together, the two axes can reflect a total of 82.97% of the species differences.

### 3.6. Analysis of Flavor Compounds and Their Correlation with Bacterial Communities

According to the above results, inoculation with *L. plantarum* L1 resulted in the most significant improvement in the quality of *Zhayu* products. Therefore, the difference in volatile flavor compounds between the CK and L1 groups was detected after 4 days of fermentation. In general, 80 volatiles were identified in *Zhayu* samples, including 14 alcohols, 1 aldehyde, 11 ketones, 14 acids, 16 esters, 10 hydrocarbons, and 14 other compounds (Appendix A). Principal component analysis (PCA) showed that volatile compounds were significantly different between the CK and L1 groups (Figure 9). A heat map was created to reflect the difference in volatile flavor compounds between the two groups (Figure 10). Additionally, the VIP values of core differential flavor substances were further analyzed using the OPLS-DA model, and 26 compounds had VIP values above 1.0. The top 10 differential flavor substances were 2,3-butanediol, dihydro-3,5-dimethyl-2(3H)-furanone, 2-heptanol, 1-methylbutyl acetate, 2-heptanone, 1-methylcyclooctanol, 4-methyl-1-pentanol, L-lactic acid, 4-methyl-pentanoic acid ethyl ester, and 2-pentylfuran (Figure 11). A Pearson correlation analysis was performed to investigate the relevance between the bacterial community and volatile flavor compounds (Figure 12). In this research, the volatile compounds of *Zhayu* products were significantly different from other fermented fish products, such as *Suanyu.* Zeng et al. [19] detected higher levels of ethanol and acetic acid in inoculated LAB. An et al. [5] discovered that the addition of LAB increased the contents of terpenoids, acids, esters, and S-containing compounds in *Zhayu* while decreasing the relative levels of 1-octen-3-ol. The individual flavor of *Zhayu* may be due to the addition of red yeast rice and the drying process.

Many alcohol compounds were reduced in the L1 group, such as 2,3-butanediol, 5-methyl-2-Heptanol, 1-propanol, and 3-methylthiopropanol, and the abundance of 2,3-butanediol was significantly reduced in the L1 group. Similar results were observed in *Suanyu* inoculated with *Lactobacillus plantarum*, which also showed a decrease in alcohol levels [11]. It was reported that the production of 2,3-butanediol was mainly related to bacteria, including *Klebsiella*, *Bacillus*, *Enterobacter*, *Pseudomonad,* and *Serratia* [38]. In our study, inoculation fermentation resulted in the dominance of the *L. plantarum* genus and observably reduced the abundance of *Klebsiella and Enterobacter* (Figure 7a and Figure 4). The Pearson correlation analysis in this study also showed that 2,3-butanediol was negatively correlated with *Lactiplantibacillus* and significantly positively correlated with *Panotoea*, *Klebsiella*, *Cronobacter*, *Staphylococcus,* and *Kocuria* (Figure 12). However, (-)-trans-myrtanol in the L1 group was 12.7 times more abundant than in the CK group. (-)-trans-Myrtanol is a natural component with an inhibiting ability on *Klebsiella pneumoniae*, which is commonly found in some essential oils [39]. It was rarely reported in the existing research on fermented fish. Moreover, 2,3-butanediol is often studied as a characteristic substance in white spirit [40]. The taste of wine in the group inoculated with *L. plantarum* decreased significantly.

Most of the acid substances in the *Zhayu* samples showed no significant difference between the two groups, except octanoic acid, which was five times more abundant in the L1 group and positively correlated with *Lactiplantibacillus.* Octanoic acid (fruit acidity) is a long-chain fatty acid that originates from lipolysis and amino acid degradation during fermentation [41]. Octanoic acid has been linked to the desirable fruity flavor of chocolate when inoculated with *Lactiplantibacillus plantarum* [42,43].

As for esters, the abundance of ethyl 2-methylbutyrate, ethyl phenylacetate, and ethyl isovalerate compared favorably to the CK group but was lower in the L1 group. Methyl bromoacetate and phthalate were more abundant in L1 group. Esters usually provide the fruity aroma for fermented products, such as dry-fermented sausages and fermented mandarin fish [44,45]. The Pearson correlation analysis in this study showed that ethyl 2-methylbutyrate and ethyl phenylacetate were significantly negatively correlated with *Lactiplantibacillus* (Figure 12).

In the other classes, the abundance of 2-pentylfuran and 2-butyl-3-methylpyrazine had increased by 3- and 9-fold in the L1 group compared with the CK group. The 2-pentylfuran was positively correlated with *Lactiplantibacillus*. Heterocyclic aromatic substances, including furan and pyrazine, have been identified as the key flavor substances in many fermented foods [46,47]. It is widely believed that furan and pyrazine are generated through Strecker degradation of amino acids and reductones under heating conditions [48]. While furan and pyrazine were also accumulated in fish products fermented at room temperature [46], the biosynthetic pathway of pyrazines has been studied in some microorganisms. Bacteria have the ability to catalyze reactions between amino acids and sugars during fermentation, resulting in the production of pyrazine [49]. During the fermentation of soybeans, *Bacillus* is capable of producing the volatile component pyrazine [50]. As the *Zhayu* sample of group L1 was fermented under 32 °C, it is probable that *L. planturem* L1 had the ability to bio-transform amino acids into pyrazines. In conclusion, each ingredient may, to varying degrees, contribute to the complex aromatic character of *Zhayu*’s acceptable aroma.

## 4. Conclusions

The findings demonstrated that, when compared to spontaneous fermentation, inoculation fermentation increased the hardness and TCA-soluble peptides of *Zhayu* by about 1.25- and 1.3-fold, respectively, reduced pH by approximately 18% and TBARS values by 45%, and prevented the growth of *Staphylococcus aureus* and *Enterobacteriaceae*. Inoculating the group with *L. plantarum* led to the best outcomes. We found that the natural fermentation samples were richer than the inoculated LAB fermentation samples in bacterial colony composition and flavor composition because the fermentation with LAB reduced the abundance of spoilage bacteria and reduced the production of unpleasant odors. In the inoculation and fermentation group, pyrazine and furan flavor substances were more abundant, and unsaturated alcohols such as 2,3-butanediol were less abundant. At the same time, the correlation between bacterial communities and volatile flavor compounds showed that the main microbial genera correlated with volatile flavor compounds are *Lactiplantibacillus*, *Pantoea*, *Klebsiella*, *Cronobacter*, *Pediococcus*, *Weissella*, *Staphylococcus*, and *Kosakonia*. Of these, *Lactiplantibacillus* was positively correlated with only five substances, 4-methyl-1-pentanol, 2-pentylfuran, 4-hydroxy-2-butanone, isohexyl pentyl-oxalic acid, ester, and octanoic acid, and negatively correlated with all other substances, while the other strains were exactly the opposite. It has been demonstrated that *L. plantarum* can improve the flavor and safety of fish. However, a reduction in colony diversity is not always a positive result. It may cause the reduction in the diversity of volatile flavor, leading to flavor homogeneity. Perhaps fermentation with mixed culture or step fermentation could be used for further flavor optimization. Further studies should be devoted to the mechanisms of influence between microbial diversity (bacteria and fungi) and flavor substances. These results will help to offer a theoretical basis for isolating and screening microbial strains for producing safe and high-quality fermented *Zhayu* products.

## Figures and Tables

**Figure 1 foods-12-01768-f001:**
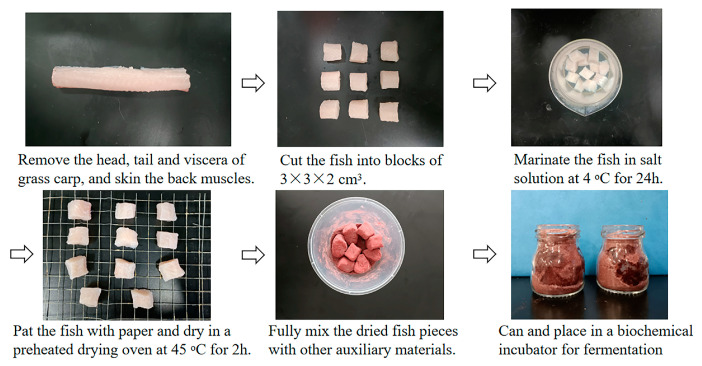
Preparation of *Zhayu* samples.

**Figure 2 foods-12-01768-f002:**
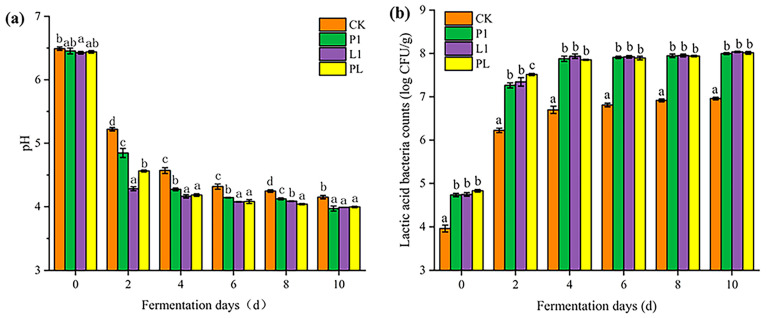
Changes in pH (**a**) and LAB (**b**) counts of *Zhayu* samples during fermentation. The four groups are CK (without starter cultures), P1 (inoculated with *P. pentosaceus* P1), L1 (inoculated with *L. plantarum* L1), and PL (inoculated with the 1:1 mixture of *P. pentosaceus* P1 and *L. plantarum* L1). Error bars show standard deviations. ^a–d^ Values in the same day with different letters were significantly different (*p* < 0.05).

**Figure 3 foods-12-01768-f003:**
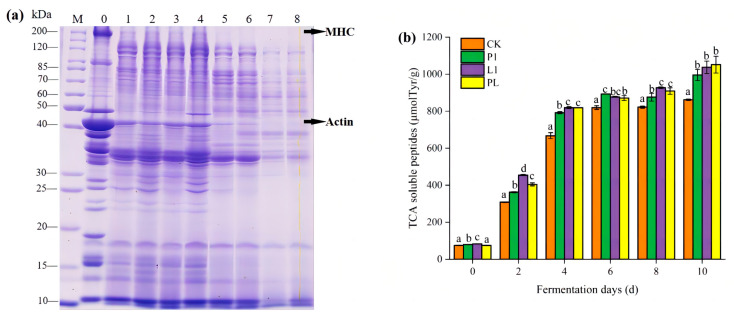
(**a**) SDS-PAGE analysis of *Zhayu* during processing. M: Marker; 0: *Zhayu* sample before fermentation. Swimming lanes 1~4 are CK (without starter cultures), P1 (inoculated with *P. pentosaceus* P1), L1 (inoculated with *L. plantarum* L1), and PL (inoculated with the 1:1 mixture of *P. pentosaceus* P1 and *L. plantarum* L1) at 2 days of fermentation. Swimming lanes 5~8 are CK (without starter cultures), P1 (inoculated with *P. pentosaceus* P1), L1 (inoculated with *L. plantarum* L1), and PL (inoculated with the 1:1 mixture of *P. pentosaceus* P1 and *L. plantarum* L1) at 4 days of fermentation. (**b**) Changes in TCA-soluble peptides of *Zhayu* samples during fermentation. The four groups are CK (without starter cultures), P1 (inoculated with *P. pentosaceus* P1), L1 (inoculated with *L. plantarum* L1), and PL (inoculated with the 1:1 mixture of *P. pentosaceus* P1 and *L. plantarum* L1). ^a–d^ Values in the same day with different letters were significantly different (*p* < 0.05).

**Figure 4 foods-12-01768-f004:**
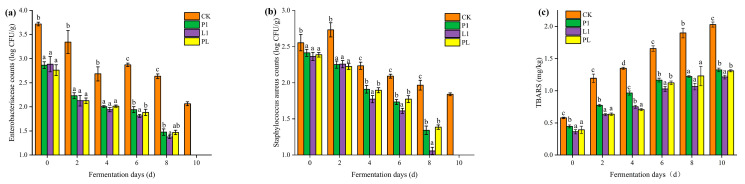
Changes in (**a**) *Enterobacteriaceae* counts, (**b**) *Staphylococcus aureus* counts, and (**c**) TBARS of *Zhayu* samples during fermentation. The four groups are CK (without starter cultures), P1 (inoculated with *P. pentosaceus* P1), L1 (inoculated with *L. plantarum* L1), and PL (inoculated with the 1:1 mixture of *P. pentosaceus* P1 and *L. plantarum* L1). ^a–d^ Values in the same day with different letters were significantly different (*p* < 0.05).

**Figure 5 foods-12-01768-f005:**
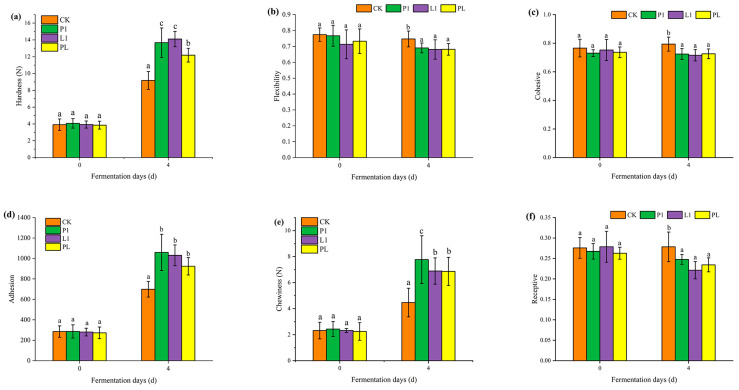
Texture parameters of *Zhayu* during fermentation: (**a**) Hardness; (**b**) Flexibility; (**c**) Cohesive; (**d**) Adhesssion; (**e**) Chewiness; and (**f**) Receptive. The four groups are CK (without starter cultures), P1 (inoculated with *P. pentosaceus* P1), L1 (inoculated with *L. plantarum* L1), and PL (inoculated with the 1:1 mixture of *P. pentosaceus* P1 and *L. plantarum* L1). ^a–c^ Values in the same day with different letters were significantly different (*p* < 0.05).

**Figure 6 foods-12-01768-f006:**
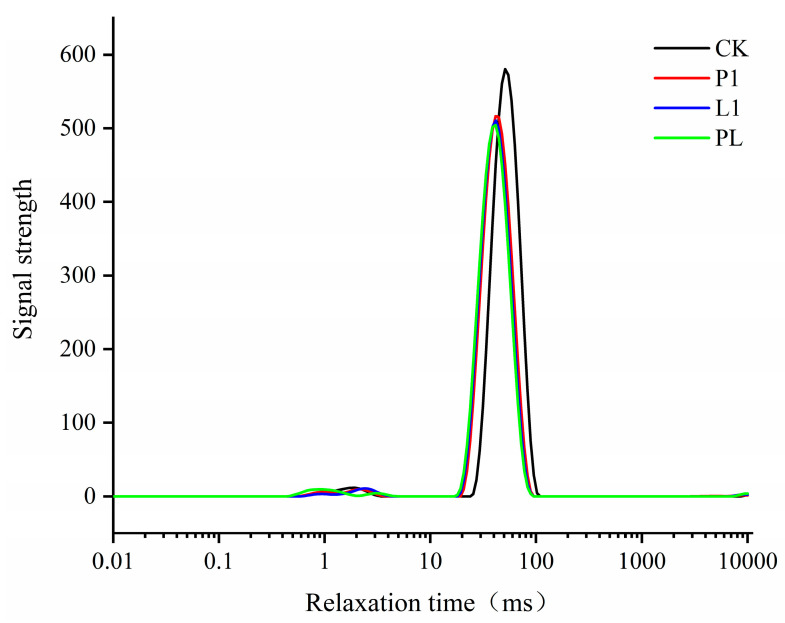
Changes in distributions of T2 relaxation time of *Zhayu* (A). The four groups are CK (without starter cultures), P1 (inoculated with *P. pentosaceus* P1), L1 (inoculated with *L. plantarum* L1), and PL (inoculated with the 1:1 mixture of *P. pentosaceus* P1 and *L. plantarum* L1). The T22 (10~100 ms) peak of CK group shifted to the right.

**Figure 7 foods-12-01768-f007:**
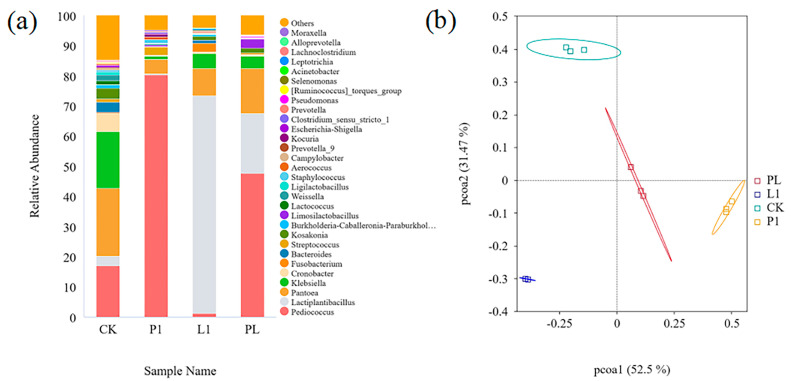
(**a**) Relative abundance of bacterial community proportions at genus level (top 30) and (**b**) PCoA analysis based on Bray−Curtis of *Zhayu* samples at 4 days of fermentation. The four groups are CK (without starter cultures), P1 (inoculated with *P. pentosaceus* P1), L1 (inoculated with *L. plantarum* L1), and PL (inoculated with the 1:1 mixture of *P. pentosaceus* P1 and *L. plantarum* L1).

**Figure 8 foods-12-01768-f008:**
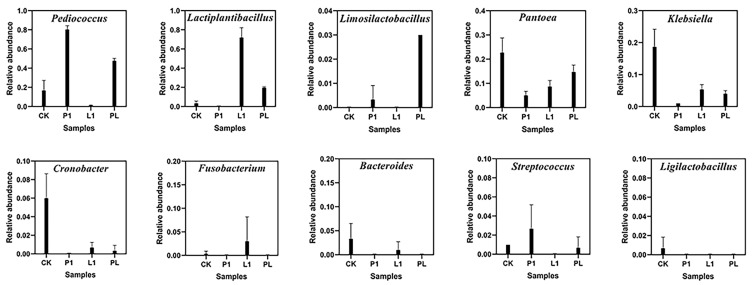
Relative abundance of bacterial community proportions of samples at 4 days of fermentation (there are *Pediococcus*, *Lactiplantibacillus*, *Limosilactobacillus*, *Pantoea*, *Klebsiella*, *Cronobacter*, *Fusobacterium, Bacteroides, Streptococcus*, and *Ligilactobacillus*). Error bars show standard deviations.

**Figure 9 foods-12-01768-f009:**
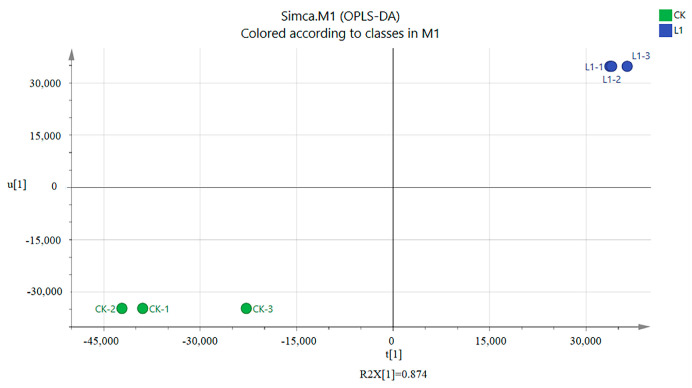
PCA based on OPLS−DA indicated the difference in volatile flavor compounds between the CK (without starter cultures) and L1 (inoculated with *L. plantarum* L1) groups at 4 days of fermentation of *Zhayu*. CK−1, CK−2, and CK−3 are the three replicates of the CK group and L1−1, L1−2, and L1−3 are the three replicates of the L1 group. The farther they are, the greater the difference.

**Figure 10 foods-12-01768-f010:**
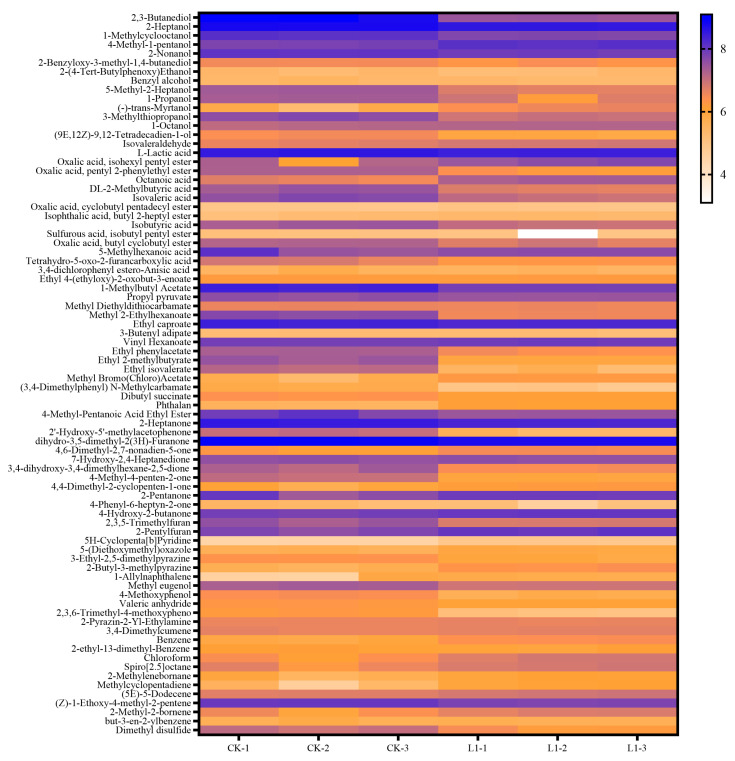
Heatmap visualization of volatile flavor compound substances in the CK (without starter cultures) and L1 (inoculated with *L. plantarum* L1) groups of *Zhayu* at 4 days of fermentation. The relative intensities rise from low (orange) to high (blue). CK−1, CK−2, and CK−3 are the three replicates of the CK group and L1−1, L1−2, and L1-3 are the three replicates of the L1 group.

**Figure 11 foods-12-01768-f011:**
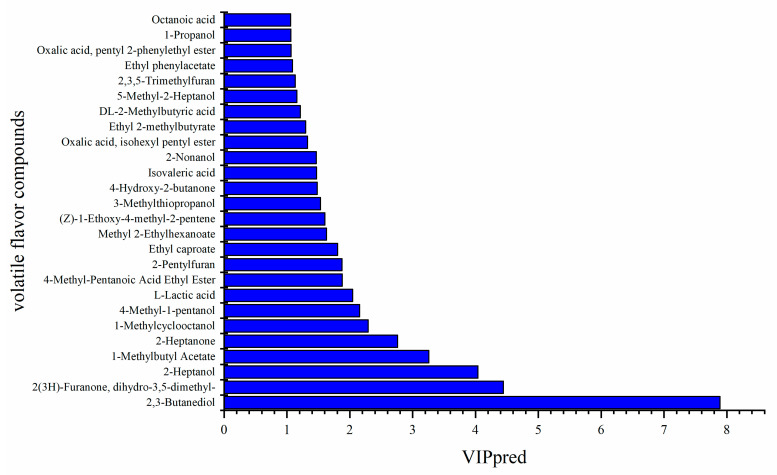
VIP values that were over 1.0 of the volatile flavor compounds predicted by the OPLS model in the CK (without starter cultures) and L1 (inoculated with *L. plantarum* L1) groups at 4 days of fermentation of *Zhayu*. The farther they are, the greater the difference.

**Figure 12 foods-12-01768-f012:**
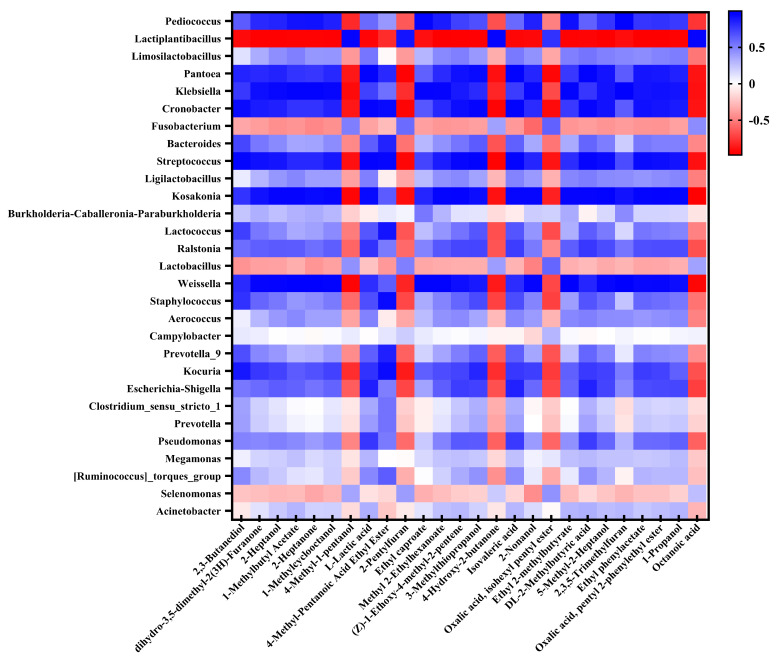
Correlation map of volatile flavor compounds with microorganisms of *Zhayu* at 4 days of fermentation. Each square indicates the Person’s correlation coefficient values (r). Positive (0 < r < 1) and negative (−1 < r < 0) correlations are indicated in blue and red, respectively.

**Table 1 foods-12-01768-t001:** Analysis of the alpha diversity index of different samples at 4 days of fermentation. The four groups are CK (without starter cultures), P1 (inoculated with *P. pentosaceus* P1), L1 (inoculated with *L. plantarum* L1), and PL (inoculated with the 1:1 mixture of *P. pentosaceus* P1 and *L. plantarum* L1). The Shannon and Simpson indexes were used to reflect species diversity; the Chao1 and Ace indexes measure the abundance of species, that is the number of species; and the P-D index reflects the kinship of species within the community. They belong to α diversity indexes.

	Observed_Species	Shannon	Simpson	Chao1	ACE	PD_Whole_Tree	OTU	Goods_Coverage
CK	1065	4.347	0.878	1278.517	1312.548	509.538	2690	0.994
P1	657	1.859	0.375	786.27	858.857	232.568	1725	0.996
L1	628	1.959	0.456	780.442	888.153	236.48	1667	0.995
PL	181	2.953	0.784	196.266	204.191	39.671	313	0.999

## Data Availability

Data are contained within this article.

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
