# Peer review of "Application and Effect of Pediococcus pentosaceus and Lactiplantibacillus plantarum as Starter Cultures on Bacterial Communities and Volatile Flavor Compounds of Zhayu, a Chinese Traditional Fermented Fish Product"

_foods, 2023, doi:10.3390/foods12091768_

Round 1

Reviewer 1 Report

This present study is aimed at evaluating the impact of inoculated LAB species on some parameters of fermented fish.

The study is interesting however some improvements are necessary.

- Abstract: specify how much increased or decreased analyzed parameters (lines No. 20, 25, and 27).

- line 76: add the Latin name of the fish.

- line 95: add citation (our previous study).

- Material and Methods: indicate the company and country of all instruments used in this study (for example, electric oven, incubator, etc.).

- line 111: describe the method used for the evaluation of soluble peptides.

- line 178: are you sure, that 5 g of products were added into 10 ml vial? 

- line 191: How was RI calculated?

- add a reference to the Fig. 1 in the text.

- please, explain, why some analyses were performed within 10 days of fermentation, and others only after 4 days?

- please, explain the header row of Table 1. What means "shanson", "simpson", etc.

- Fig 8. - Latin names must be in italics.

- Fig. 11 - are compounds listed according to their elution order out of the column?

- Fig. 11 - check chemical titles, for example, it should be "3,4-dihydroxy-3,4-dimethylhexane-2,5-dione" instead of "34-dihydroxy-34-dimethylhexane-2,5-dione", and others.

Fig. 11 - please, explain the parameters indicated in X axis. Why there are CK1, CK2, CK3 and L1-1, L1-2, L1-3? while in the previous text, it was only CK and L1...

-line 533 - how much higher quantities? It may be 1-2 percent higher or 50 percent...

Table S1 is missing.

Latin names in the reference list must be italicized.

Reviewer 2 Report

The effect of Pediococcus pentosaceus and Lactiplantibacillus plantarum as starter cultures on bacterial communities and volatile flavor compounds of Zhayu, a Chinese traditional fermented fish product is investigated in the present manuscript.

Although it is an interesting and well-designed study, there are several issues that should be addressed.

Specifically:

Line 21: “TCA-soluble compounds” TCA abbrev. should be explained the first time it is mentioned in the manuscript

Line 21-22: “Texture profile analysis 21 were performed...” replace with “was performed”

The results presented on the abstract are scarce and there is no reference on P. pentosaceus inoculation. Authors should rewrite the Results & Discussion section of the abstract.

Line 50-51: Sentence starts with And

Line 66: “good fermentation characteristics” What does this mean?

Line 85: MRS is mentioned here for the second time (it is mentioned before in line 80) but the abbreviation is explained in the second time.

Line 87-88: Re-suspending the bacteria cell pellets in sterile saline water and adjusting the cell density to around 106 CFU/mL. This is not a grammatically right sentence. Please check English language use

Line 95: “Based on our previous study” please give the reference of your previous study

Line 100-101: “Take 100 three parallel samples randomly each time for inspection”. This is not a sentence

Line 103-104: “The pH values of Zhayu were measured according to the national standard GB 103 5009.237-2016.” There is no reference.

Why did you use this specific strains? Where are they isolated from? Are they novel? What about their safety? Authors should discuss these issues and provide the necessary information.

Line 242: “…in other studie…” should be corrected to “…in anoother study…”

Line 242: “L. plantarum showing” This is not English

Line 292-293: “and could not be detected at 10 days of fermentation”. Please report the detection limit (1 log? 2 log?) and clarify if specific limits are required in such products. Furthermore, why did you only monitor Enterobacteriacae and Staphylococcus aureus? What about other bacteria? And what about yeasts? Is there any official guide for such fermented products?

Line 297: “has showed the better antibacterial capability” has shown?? the better?? English again.. is this “better antibacterial capability” significant? what do you compare it to with? how about p value?

How do you correlate the results in Fig 4? The first 2 diagrams are the logcfu of the microbiological analyses and then the TBARS results are shown. How are these things related? Please consider presenting the data in different graphs

Line 310: Enterococcus is responsible for food poisoning but you did not count enterococcus in your samples? You could use a proper agar medium for Enterococcus enumeration. Please justify why you did not perform the analysis.

Line 326: were showed or are shown? are presented?

Line 328: there was

Line 268: was performed

Line 368-369: Please explain why you choose to sequence samples only from the 4th day

Alpha diversity results imply that naturally fermented fish consisted of more diverse microbes. Please discuss on this finding as it means that the addition of the LAB led to decreased diversity. Is this a “positive” finding?

Conclusions are messy and the authors should try to address a next-step application of their research

Reviewer 3 Report

The manuscript “Application and effect of Pediococcus pentosaceus and Lacti plantibacillus plantarum as starter cultures on bacterial communities and volatile flavor compounds of Zhayu, a Chinese traditional fermented fish product” is designed and executed well. Results are satisfactory and documented properly. I recommend a minor revision.

1. Figure 1 is not mentioned in the text. Add Fig 1 in the text MS.

2. There are too many figures in the MS. It would be better if the author can merge figure 7, 8, 9, and 10 and present it as combined figures.

3. What % of the polyacrylamide gel was used for the separation?

Round 2

Reviewer 2 Report

The manuscript has been significantly improved